# A Review of Nanofluids as Coolants for Thermal Management Systems in Fuel Cell Vehicles

**DOI:** 10.3390/nano13212861

**Published:** 2023-10-28

**Authors:** Qi Tao, Fei Zhong, Yadong Deng, Yiping Wang, Chuqi Su

**Affiliations:** 1Hubei Key Laboratory of Modern Manufacture Quality Engineering, School of Mechanical Engineering, Hubei University of Technology, Wuhan 430068, China; hg_zfxs@sina.com; 2Hubei Key Laboratory of Advanced Technology for Automotive Components, Wuhan University of Technology, Wuhan 430070, China; dengyadong@sina.com; 3Hubei Collaborative Innovation Center for Automotive Components Technology, Wuhan University of Technology, Wuhan 430070, China; wangyiping@whut.edu.cn; 4Hubei Research Center for New Energy & Intelligent Connected Vehicle, Wuhan University of Technology, Wuhan 430070, China; suchuqi@163.com

**Keywords:** nanofluids, coolant, thermal management system, fuel cell vehicles

## Abstract

With the development of high-power fuel cell vehicles, heat dissipation requirements have become increasingly stringent. Although conventional cooling techniques improve the heat dissipation capacity by increasing the fan rotating speed or radiator dimensions, high energy consumption and limited engine compartment space prevent their implementation. Moreover, the insufficient heat transfer capacity of existing coolants limits the enhancement of heat dissipation performance. Therefore, exploring novel coolants to replace traditional coolants is important. Nanofluids composed of nanoparticles and base liquids are promising alternatives, effectively improving the heat transfer capacity of the base liquid. However, challenges remain that prevent their use in fuel cell vehicles. These include issues regarding the nanofluid stability and cleaning, erosion and abrasion, thermal conductivity, and electrical conductivity. In this review, we summarize the nanofluid applications in oil-fueled, electric, and fuel cell vehicles. Subsequently, we provide a comprehensive literature review of the challenges and future research directions of nanofluids as coolants in fuel cell vehicles. This review demonstrates the potential of nanofluids as an alternative thermal management system that can facilitate transition toward a low-carbon, energy-secure economy. It will serve as a reference for researchers to focus on new areas that could drive the field forward.

## 1. Introduction

### 1.1. Nanofluid Overview

Nanofluids were first proposed by Professor Choi at Aragon National Laboratory, the United States of America in the early 1990s. They are composed of nanoparticles, metal, or polymer particles of sizes ranging from 1 nm to 100 nm and the base liquid. This is shown in Figure 1. Nanofluids have a capacity to enhance the heat dissipation performance of the base fluid. They exhibit a variety of unique thermal, optical, electrical, magnetic, and other properties, and they have hence been innovatively utilized in heat transfer engineering and applied in different fields including aerospace, energy, electromechanics, and biomedicine [1].

The heat transfer enhancement mechanism of a nanofluid involves the addition of nanoparticles to the base fluid, increasing the thermal conductivity of the mixture and thereby improving the heat transfer capacity of the base fluid. In addition, particle-to-particle and wall-to-wall collisions and the mutual motion between the particles and liquid improve the heat transfer characteristics. 

However, this is a complicated process in microchannels. Molecular dynamics have been used to analyze the heat transfer mechanism at solid–liquid interfaces. Frank [2,3] investigated the effects of surface irregularities and imperfections on thermal resistance using molecular dynamics (MD). The thermal resistance was reduced by increasing the strength of the solid–liquid interface. Ghorbanian [4] used nonequilibrium molecular dynamics to analyze the effects of the wall force field and scale. The results proved that wall effects lead to continuum solution deviations. Papanikoulaou [5] used molecular dynamics to analyze the effects of surface roughness on nanoflows. The results showed that the flow velocity near the wall was reduced with an increasing roughness depth. Liu [6] explored the transport and structural characteristics of water inside nanotubes using molecular dynamic simulations. The results indicated that thermal conductivity and shear viscosity increased as the pore size decreased. McGaughey [7] also used molecular dynamics simulations to analyze the heat transport mechanism of silica-based crystals. The simulation results showed that thermal conductivity is related to the scale and temperature of the atomic structure of silica-based crystals. Sun [8] used the colloidal probe technique to analyze the force profiles of nanoparticles and base fluids. The test results showed that the measured and calculated force curves exhibited an evident slip (10–14 nm).

Nanofluids can be used as coolants in systems with high heat-dissipation requirements. For example, nanofluid applications have been investigated in food processing, solar energy collection, nuclear reactors, and oil recovery. In food processing, nanofluids can be used to increase heat exchanger efficiency and reduce energy consumption. Nanofluids can shorten the thermal processing time and improve the nutritional content of food [9,10,11,12,13,14]. In the solar energy collecting field, nanofluids have been proven to increase the collecting efficiency of solar energy collectors, including parabolic trough solar collectors, evacuated tube solar collectors, flat plate solar collectors, and solar cookers [15,16,17,18,19,20,21,22]. This is illustrated in Figure 2. In the nuclear reactor field, nanofluids as coolants could increase the cooling capacity and maintain safe operation of nuclear reactor cooling systems [23,24,25,26,27,28,29,30,31,32,33,34,35]. Nanofluids also can be used to improve oil recovery efficiency due to their excellent heat transfer performance [36,37,38,39,40,41,42,43,44,45,46,47,48,49,50,51]. For example, nanofluid applications in oil-fueled vehicles include their use in cooling, lubrication, air conditioners, and exhaust power generation systems, as shown in Figure 2.

In 2004, the U.S. Department of Energy used nanofluids in fuel cells, and the heat dissipation capacity of cooling systems could be significantly improved [52]. Particularly, for fuel cell vehicles, nanofluids have the potential to replace existing coolants and improve the heat transfer capacity of thermal management systems. This paper will explore the challenges and future research directions of nanofluids as coolants in fuel cell vehicles.

### 1.2. Nanofluid Application in Different Types of Vehicles

#### 1.2.1. Nanofluid Application in Oil-Fueled Vehicles

Nanofluid application in oil-fueled vehicles’ cooling systems

Nanofluids can be used to improve the heat-dissipation capacity of radiators in cooling systems. Many researchers have investigated the mechanism of heat transfer enhancement in radiator tubes using different nanofluids. Table 1 presents a summary of the nanofluid applications in oil-fueled-vehicle cooling systems. 

According to the literature content in Table 1 above, nanoparticles including Al_2_O_3_, TiO_2_, CuO, SiO_2_, Fe_3_O_4_, Fe_2_O_3_, ZnO, SiC, GnP, CNC, MWCNT, CNT, and graphene have been analyzed in detail. In addition, the thermophysical properties and heat transfer performance of different types of nanofluids have been analyzed in the past. The influencing factors, including concentration, flow rate, temperature, and radiator dimensions, have been compared. Using a nanofluid instead of a conventional coolant has been demonstrated to improve the heat dissipation performance of thermal management systems [100,101,102,103,104,105,106,107].

Nanofluid applications in oil-fueled vehicles’ air conditioner systems

Sharif [108] chose a nanofluid as a lubricant to enhance the compressor performance. A SiO_2_/PAG nanofluid was selected as the nano-lubricant. A vehicle air conditioner test rig was used to measure the heat absorption and coefficient of performance (COP). The test results indicated that the maximum increase in the COP was 24% when using the SiO_2_/PAG nanofluid. Redhwan [109] prepared Al_2_O_3_-SiO_2_/PAG nanofluids and compared the heat transfer performance and tribology enhancement. Zawawi [110] used an Al_2_O_3_-SiO_2_/PAG nano-lubricant and analyzed its effect on the performance improvement of vehicle air conditioner systems through experimental tests. The test bench is illustrated in Figure 3. It was composed of a power analyzer, frequency inverter, electric motor, water bath, compressor, data logger, water heater, and condenser.

Nanofluid applications in oil-fueled vehicles’ lubrication systems

The main reason for the failure of lubricating oils at high temperatures is thermal degradation. Nanofluids can delay the oxidation and burnout of the lubricating oil, which results in a wider temperature range and higher fuel economy. Ali [111] used a thermal analyzer (TGA/DTG/DSC) to investigate the thermal degradation of lubricating oil. An AVL dynamometer was used to test the engine thermal efficiency with the Al_2_O_3_-TiO_2_/lube nanofluid. Lue [112] used a two-step method to prepare four different weight fractions of an Al_2_O_3_/lube nanofluid. The densities and viscosities of the nano-lubricant were measured at various temperatures. The test results indicated that the 1.5 wt % Al_2_O_3_/lube nanofluid was the optimal choice for future research. Esfe [113] proposed a type of MWCNT-ZnO nano-lubricant for light-duty vehicles that can be used to reduce viscosity and effectively improve the engine’s efficiency.

Nanofluid applications in oil-fueled vehicles’ exhaust power generation systems

Researchers have investigated the possible utilization of waste heat in exhaust pipes because of the large amount of waste heat produced during engine operation. Based on the thermoelectric conversion unit, waste heat can be converted into electricity. For example, Hilmin [114] prepared a TiO_2_/water nanofluid using a two-step method to increase the efficiency of conversion units in thermoelectric generators (TEGs). The experimental results demonstrated that the TiO_2_/water nanofluid effectively enhanced the conversion efficiency and output power of the TEG. Karana [115] prepared ZnO/WEG and MgO/WEG nanofluids that were used to improve the efficiency of waste heat recovery for a TEG. The influencing factors included the Reynolds number, total TEG area, nanofluid concentration, and inlet temperature. According to the analysis results, the total TEG area was reduced by 33% with a 1% MgO/WEG nanofluid.

#### 1.2.2. Nanofluid Application in Electric Vehicles

Batteries are the most important power source for electric vehicles. However, maintaining an ordinary operating temperature using a thermal management system is crucial. Nanofluids have been used as a coolant to replace existing coolants, and their performance in improving heat-dissipation capacity has been analyzed. For example, Chen [116] constructed a battery heating module with a pipe and selected a TiO_2_-CLPHP nanofluid as the coolant. The heating performances of different strategies were investigated. The test results indicated that the 2% TiO_2_/CLPHP nanofluid exhibited the best heat transfer performance. Wu [117] used a Cu/H_2_O nanofluid with a lattice Boltzmann (LB) model to simulate natural convection in a battery. The problem of a uniformly distributed heat boundary was validated to ensure the accuracy of the model. The results proved that the Cu/H_2_O nanofluid could improve the cooling performance and reduce the temperature difference in a battery thermal management system. Abdelkareem [118] compared different heat-dissipation capacity-enhancement methods for batteries, including air cooling, liquid cooling, heat pipes, and phase change material and found that the nanofluid effectively enhanced the heat-dissipation capacity. Guo [119] proposed a type of battery cooling system with a liquid cold plate and analyzed the different factors influencing the heat-dissipation performance. The simulation results showed that the maximum temperature and temperature differences were separately reduced. Adding nanoparticles (Al, Cu, and Ag) to the base fluid could effectively improve the heat transfer performance. Kiani [120] designed a hybrid cooling system with nanofluid as the coolant. The results indicated that the nanofluid ensured battery safety under stressful conditions. Moreover, many researchers have analyzed the heat-dissipation improvement in battery thermal management systems using nanofluids as coolant [121,122,123,124,125,126,127,128,129,130].

#### 1.2.3. Nanofluid Application in Fuel Cell Vehicles

In 2004, the U.S. Department of Energy first selected nanofluids as coolants for thermal management systems in fuel cell vehicles. The preparation cost of the nanofluid can be reduced and is close to that of the traditional coolants used in oil-fueled vehicles. Nanofluids can be used as coolants to reduce the radiator dimensions. They can also effectively improve the heat-dissipation capacity of cooling systems [52]. Many researchers have conducted preliminary studies on the application of nanofluids in fuel cell vehicles. For example, Bargal [131] analyzed the research progress on nanofluids in a PEMFC and compared different cooling methods for thermal management systems in fuel cell vehicles. Islam [132] discussed the application of nanofluids as coolants to improve the heat-dissipation capacity of thermal management systems. The results proved that nanofluids can reduce the radiator dimensions and enhance the heat-dissipation capacity. Islam [133] analyzed a 0.5% ZnO/WEG nanofluid as a coolant to improve the heat-dissipation capacity of thermal management systems. The results showed that the frontal area of the radiator could be reduced by 27% compared to when EG/water was used as a coolant. Islam [134] investigated the electrical conductivity and heat transfer coefficient of TiO_2_/WEG nanofluid at 0.05–0.5%. The experimental results indicated that the thermal conductivity coefficient increased by 10% with the 0.5% TiO_2_/WEG nanofluid. Zakaria [135] investigated the heat-dissipation performance of 0.1% and 0.5% Al_2_O_3_/WEG nanofluids in PEMFCs. The experimental results showed that the heat-dissipation power increased by 13.87% with a 0.5% Al_2_O_3_/WEG nanofluid. Zakaria [136] analyzed the heat transfer performance of 0.1% and 0.5% Al_2_O_3_/WEG nanofluids with different base fluid ratios. The test results indicated that the heat transfer performance of the 0.1% Al_2_O_3_/WEG nanofluid at a 60:40 ratio was better than that at a 50:50 ratio. Zakaria [137] also analyzed the ratio of thermal conductivities and electrical conductivities of 0.1%, 0.3%, and 0.5% Al_2_O_3_ nanofluids. The testing results showed that the thermal conductivity was reduced when the ethylene glycol proportion was increased. They also showed that the electrical conductivity was reduced when the ethylene glycol proportion was increased. In order to assess the significance of Al_2_O_3_ nanofluids in an electrically active thermal device, the thermoelectrical conductivity (TEC) ratio was established.

The aforementioned research studies were mainly based on metallic nanoparticles. Boron nitride (BN) is an insulating ceramic with high stability, high thermal conductivity, and good chemical inertness. For example, Ilhan [138] prepared a nanofluid from boron nitride (BN) and analyzed its heat transfer performance. The test results proved that BN nanofluid has considerable potential for use as a coolant in fuel cell vehicles.

## 2. Nanofluid Application in Thermal Management Systems of Fuel Cell Vehicles

### 2.1. Thermal Management System Structure

The thermal management system of a fuel cell vehicle (FCV) comprises heat dissipation and air conditioning systems. The heat-dissipation system includes heat-generating (fuel cell stack, DC/DC converter, motor, motor controller, and air compressor) and heat-dissipation (radiator, water pump, fan, and expansion tank) components, as shown in Figure 4.

A proton-exchange membrane fuel cell (PEMFC) engine comprises an air supply system, control system, cooling system, fuel cell stack, and fuel supply system. Among these, the fuel cell stack is composed of a single PEMFC, which converts chemical into electrical energy. The cooling system of the PEMFC includes a radiator, fan, pump, intercooler, thermostat, and water tank. The gas temperature at the cathode inlet is adjusted using the coolant temperature and flow rate inside the PEMFC. The coolant can be circulated in a cooling loop, and heat is transferred from the interior of the fuel cell. Figure 5 shows a schematic diagram of a PEMFC engine. The systems and components work together to complete the normal operation of the fuel cell engine.

### 2.2. Cooling Performance Improvement of Thermal Management System

#### 2.2.1. Heat-Production Components

Many researchers have investigated methods for optimizing the cooling techniques of heat-dissipation components to maintain the ordinary operating temperature of PEMFCs in fuel cell vehicles. As mentioned earlier, the heat-production components of fuel cell vehicles include fuel cells, power batteries, motors, motor controllers, air compressors, and DC/DC converters.

Cooling techniques for fuel cells

According to the different output powers of fuel cells, their cooling methods can be classified as cathode air, separate airflow, phase change, and liquid cooling. The results are presented in Table 2. Presently, the output power of fuel cell engines in vehicles is higher than 10 kW. Liquid cooling is the primary cooling technique for fuel cell engines.

The heat-dissipation performance can be improved by optimizing the internal flow channel structure and adjusting the fuel cell control strategy. The design optimization of fuel cells includes the use of flow channels, bipolar plates, and polymer film cooling channels. The structure of the flow channel includes serpentine, parallel, wave, and spiral flow channels, as shown in Figure 6. The efficiency of fuel cells and different cooling fields was analyzed using a numerical method. It was found that the nearest route to the entrance had a worse fluid distribution than the farthest route. Optimization of the flow channel inside the FCs effectively improved the temperature uniformity and heat exchange rate. Thus, this can enhance a fuel cell’s stability, durability, heat-dissipation capacity, and lifetime [141,142,143]. 

In addition, a reasonable control strategy can precisely control the rotational speed of the pump and fan of the cooling system under different operating conditions, ensuring that the fuel cell engines are in the normal operating temperature range to prevent overheating or overcooling. This is helpful in enhancing the operating efficiency of fuel cells [144,145].

Cooling techniques for batteries

The cooling techniques for batteries include air and liquid cooling. Air cooling can be classified into natural or forced convection. The forced air-cooling technique has the advantages of a simple structure and low manufacturing cost. Moreover, it can increase the heat transfer coefficient on the air side and improve the heat-dissipation capacity relative to natural convection [146]. Figure 7 shows the schematic of forced air cooling. The battery system is composed of 72 battery cells (270 V and 1400 Wh). The coolant passage between the battery cells is 3 mm, which needs to dissipate a heat flux of 245 W/m^2^.

Currently, many researchers have investigated the effect of different air-cooling structures and arrangements on cooling performance using theoretical analyses and numerical simulations. For example, the effect of a conical structure on the cooling performance of a battery was investigated. Considering the influencing factors, including the discharge rate, fin arrangement, inlet air temperature, and inlet air flow rate, a prismatic air-cooled module and an aluminum-finned radiator were designed [146,147,148,149,150]. However, the liquid-cooling technique is more effective than the air-cooling technique.

Cooling techniques for other components

Cooling techniques for the motor and motor controllers include natural, liquid, air, and oil cooling. The cost of air cooling is lower than that of other cooling techniques. However, the heat-dissipation capacity is not as high as that of liquid cooling. The cooling techniques for air compressors include air and liquid cooling. The former cooling method can be improved by optimizing the structure of the air compressor. This cooling method can be improved by enhancing the thermophysical properties of the coolant. Liquid cooling has a better heat-dissipation performance than air cooling. Liquid cooling is also used for DC/DC converters; the cooling can be improved by enhancing the thermophysical properties of the coolant or by improving the structure of the DC/DC converter. Table 3 summarizes the improvement methods for the different heat-production components. Liquid cooling was found to be the best cooling method for all the heat-dissipation components.

#### 2.2.2. Heat-Dissipation Components

The heat-dissipation components of thermal management systems include radiators, pumps, and fans. For the radiator, heat-dissipation-capacity improvement is mainly achieved by structural and material optimization, where the structure includes fin and flat tube structures [151,152,153,154,155,156,157,158,159,160,161,162,163,164,165,166,167]. Many researchers have investigated the structure and material optimization for radiators. For example, Habibian [151] investigated the structure of a flat tube and fin of a radiator using numerical analysis. The research results indicated that the louver fin exhibited the best heat-dissipation performance, as shown in Figure 8. Due to the fact that an automotive vehicle radiator is composed of hundreds of fins and a few tubes, the heat transfer performance and pressure drop can be analyzed in one fin. Vaisi [152] investigated the air-flow pressure drop and air-side heat transfer characteristics of a louver fin. The analysis results indicated that the heat-dissipation performance of the radiator was affected by its dimensions and Reynolds number.

Pumps and fans are the main heat-dissipation components of thermal management systems, and they can be improved by optimizing the structure and control strategy. For example, Liu [168] used a computational fluid dynamics (CFD) method to analyze the heat-dissipation capacity of a seven-blade fan. They considered influential factors, including the blade number, air velocity, and pressure, in the heat-dissipation performance of the fan.

In summary, the radiator, pump, and fan were improved via structural optimization. Structural optimization is more complex and expensive than improving the thermophysical properties of coolants. According to the above literature, there is considerable potential for investigating nanofluids as coolants to improve the heat-dissipation performance of thermal management systems in fuel cell vehicles.

## 3. Nanofluids as Coolants in Fuel Cell Vehicles

### 3.1. Nanofluid Thermophysical Properties

The thermophysical properties of nanofluids, including density, specific heat, and thermal conductivity, have been extensively investigated. The equations for the thermophysical properties are as follows [132].

Nanofluid density can be calculated as shown in Equation (1):(1)$$\rho_{nf}=(1-\varphi )\rho_{bf}+\varphi \rho_{p}$$
where $\rho_{nf}$ is the nanofluid density, $\rho_{bf}$ is the base fluid density, $\rho_{p}$ is the nanoparticle density, and φ is the nanofluid volume fraction.

Nanofluid specific heat can be calculated as shown in Equation (2):(2)$$C_{nf}=(1-\varphi )C_{bf}+\varphi C_{p}$$
where *C_nf_* is the nanofluid specific heat, *C_bf_* is the base fluid specific heat, *C_p_* is the nanoparticle specific heat, and *φ* is the nanofluid volume fraction.

Nanofluid thermal conductivity was calculated by Hamilton & Crosser as shown in the following Equation (3):(3)$$\frac{k_{nf}}{k_{bf}}=\frac{k_{p}+\left(n-1\right)k_{bf}+\left(n-1\right)\left(k_{p}-k_{bf}\right)\varphi }{k_{p}+\left(n-1\right)k_{bf}-\left(k_{p}-k_{bf}\right)\varphi }$$
where *k_nf_* is the nanofluid thermal conductivity, *k_bf_* is the base fluid thermal conductivity, *k_p_* is the nanoparticle thermal conductivity, *φ* is the nanofluid volume fraction, and n is the shape factor.

According to the literature above, ZnO, Al_2_O_3_, TiO_2_, and BN nanoparticles can be added into the base fluid to prepare a new coolant for fuel cell vehicles. The thermophysical parameters of four nanofluids can be seen in Table 4 below. Hence, nanofluids’ thermophysical properties can be calculated by using Equations (1)–(3).

The calculation results are presented in Figure 9 below. The nanofluid density increases with an increasing volume fraction. At the same volume fraction, the density of ZnO nanofluid > TiO_2_ nanofluid > Al_2_O_3_ nanofluid > BN nanofluid. The specific heat of the nanofluid decreases as the volume fraction increases. The specific heat of BN nanofluid > Al_2_O_3_ nanofluid > TiO_2_ nanofluid > ZnO nanofluid at the same volume fraction. The nanofluid thermal conductivity also increases as the volume fraction increases. At the same volume fraction, the thermal conductivity of BN nanofluid > ZnO nanofluid > Al_2_O_3_ nanofluid > TiO_2_ nanofluid. 

### 3.2. Nanofluid Thermal Conductivity Model

Thermal conductivity is one of the most important thermophysical properties used to calculate the heat-dissipation power of a nanofluid. It is affected by different factors, including the nanoparticles’ thermal conductivity, concentration (volume fraction or mass fraction), shape, and size, as well as the base fluid’s thermal conductivity. Several classical thermal conductivity models have been considered. For example, the Maxwell model, which is a thermal conductivity model for solid–liquid and two-phase coolants, was the first proposed. It is a static model for determining the thermal conductivity of monodisperse, low-volume-fraction, and spherical nanoparticles, as shown in Equation (4) [169]. Subsequently, many researchers proposed thermal conductivity models, including the Braggeman model, Yamada and Ota model, Wasp model, and Hamilton and Crosser model [170]. The Braggeman model is a type of thermal conductivity model suitable for solid–liquid suspensions with low concentrations. The calculation results obtained using the Braggeman model were close to those of the Maxwell model [133]. The Hamilton and Crosser model is an extension of the Maxwell model that introduces sphericity to the consideration. The thermal conductivity of solid–liquid suspensions with different nanoparticle shapes were calculated [171]. The Yamada and Ota model is a modified version of the Hamilton and Crosser model that replaced the (n − 1) factor with the parameter k [172]. Davis [173] proposed a thermal conductivity model for nanofluids containing spherical particles. It can be used to calculate the heat transfer rate between solid particles and the thermal conductivity from the nanoparticle to the base fluid.
(4)$$\frac{k_{nf}}{k_{bf}}=\frac{k_{p}+2k_{bf}+2\left(k_{p}-k_{bf}\right)\varphi }{k_{p}+2k_{bf}-2\left(k_{p}-k_{bf}\right)\varphi }$$
where *K_nf_* is the effective thermal conductivity of the solid–liquid suspension, *K_p_* is the thermal conductivity of the nanoparticles, *K_bf_* is the thermal conductivity of the base fluid, and *φ* is the volume fraction of the nanoparticles.

Bruggeman [132] proposed a thermal conductivity model for a nanofluid in which the particles are distributed randomly. If the volume fraction of the nanofluid is low, the thermal conductivity is close to the experimental results. This model can be seen in Equation (5):(5)$$k_{nf}=\frac{1}{4}\left[3\left(\varphi -1\right)\frac{k_{p}}{k_{b}}+\left(2-3\varphi \right)+\frac{1}{4}\sqrt{\Delta }\right]$$
(6)$$\Delta =\left[3{\left(\varphi -1\right)}^{2}{\left(k_{p}/k_{b}\right)}^{2}+{\left(2-3\varphi \right)}^{2}+2\left(2+9\varphi -9\varphi^{2}\right)\left(k_{p}/k_{b}\right)\right]$$
where *k_p_* is the thermal conductivity of the nanoparticles, *φ* is the volume fraction of the nanofluid, and *k_b_* is the thermal conductivity of the base fluid.

The Hamilton and Crosser model is extended from the Maxwell model by introducing sphericity, which allows the calculation of the thermal conductivity of solid–liquid suspensions formed by spherical or non-spherical nanoparticles [171]. This model can be seen in Equation (7):(7)$$\frac{k_{nf}}{k_{bf}}=\frac{k_{p}+\left(n-1\right)k_{bf}+\left(n-1\right)\left(k_{p}-k_{bf}\right)\varphi }{k_{p}+\left(n-1\right)k_{bf}-\left(k_{p}-k_{bf}\right)\varphi }$$
where *n* is the shape factor:(8)n=3/ψ
where *Ψ* is the sphericity degree, which is the ratio between a sphere’s surface area and a particle’s surface area at the same volume.

Yamada and Ota modified the Hamilton and Crosser model by replacing the (*n* − 1) factor with the parameter k in the equation [172]. This can be seen in Equation (10):(9)$$\frac{k_{nf}}{k_{bf}}=\frac{k_{p}+kk_{bf}+k\left(k_{p}-k_{bf}\right)\varphi }{k_{p}+kk_{bf}-\left(k_{p}-k_{bf}\right)\varphi }$$
where *k_bf_* is the base fluid thermal conductivity, *φ* is the nanofluid volume fraction, and *k_p_* is the thermal conductivity of the nanoparticles.
(10)$$k=2\varphi^{-0.2}$$

When a nanoparticle is cylindrical, the expression is as follows:(11)$$k=2\varphi^{0.2}\frac{L}{d}$$
where *L* is the nanoparticle length, and *d* is the nanoparticle diameter.

Davis proposed a model of thermal conductivity that contains spherical particles. However, Davis’s model is only suitable for calculating the thermal conductivity of spherical particles and a low volume fraction [173]. It can be seen in Equation (12):(12)$$\frac{k_{nf}}{k_{bf}}=1+\frac{3\left(1-\alpha \right)\varphi }{\left(1+2\alpha \right)-\left(1-\alpha \right)\varphi }\left[\varphi +f\left(\alpha \right)\varphi^{2}+O\left(\varphi^{3}\right)\right]$$
(13)$$f\left(\alpha \right)=\sum_{p=6}^{{}^{\infty }}\left[\left(B_{p}-3A_{p}\right)/\left(p-3\right)2^{p-3}\right]$$
where *A_p_* and *B_p_* are the constants related to parameters *α* and *p*.

However, predicting nanofluid thermal conductivity accurately using the aforementioned thermal conductivity models is difficult, and many researchers have proposed a theory of adsorption layer thermal conductivity, which refers to the liquid layer formed around solid nanoparticles. The thermal conductivity of the adsorption layer is lower than that of the nanoparticles and higher than that of the base fluid. The thermodynamic properties of the nanoparticles can be transferred to the base liquid, which is reflected in the change in the nanofluid’s thermal conductivity [174]. However, the adsorption layer of a nanofluid is difficult to measure and observe from a microscopic perspective, which is a concept derived from theoretical analyses by researchers. Figure 10 illustrates a schematic of nanoparticles and adsorption layers. 

The thermal conductivity curve for the adsorption layer includes linear and exponential distribution curves. The model for the linear distribution law of the thermal conductivity within the adsorption layer is shown in Equation (14):(14)$$k_{l}(r)=\frac{k_{bf}-k_{p}}{t}r+\frac{k_{bf}(r_{p}+t)-k_{bf}r_{p}}{t}$$
where *k_l_(r)* is the thermal conductivity within the adsorption layer, *k_bf_* is the base fluid thermal conductivity, *r_p_* is the nanoparticle size, t is the adsorption layer thickness, *k_p_* is the nanoparticle thermal conductivity, and *k_l_* is average thermal conductivity of the nanofluid adsorption layer.

However, there is no unified mathematical equation for calculating the thermal conductivity of nanofluids in fuel cell vehicles. The adsorption layer theory must be used to calculate the nanofluid thermal conductivity more accurately.

### 3.3. Single and Hybrid Nanofluids

In recent years, two or more nanoparticles have been mixed to prepare multi-mixed nanofluids. Subsequently, the heat-dissipation performances of PEMFCs were investigated. For example, Huang [175] investigated a hybrid nanofluid of Al_2_O_3_ and MWCNTs using experimental analysis. The results showed that heat transfer coefficient of the hybrid nanofluid was slightly higher than that of the Al_2_O_3_ nanofluid at the same flow velocity. The hybrid nanofluid had the highest heat transfer coefficient at a given pumping power. Arif [176] investigated a water-based ternary hybrid nanofluid that included an Al_2_O_3_ nanofluid, a CNT nanofluid, and a graphene nanofluid. The analysis results indicated that heat transfer rate of the hybrid nanofluid was increased by 33.67%. Tlili [177] prepared a hybrid nanofluid that was composed of Ni/Al_2_O_3_ nanoparticles and C_2_H_6_O_2_ as the base fluid and analyzed the heat transfer performance of the hybrid nanofluid. Devi [178] used a numerical method to compare the heat transfer rate between a Cu nanofluid and a Cu/Al_2_O_3_ nanofluid. The results proved that the heat transfer rate of the Cu/Al_2_O_3_ nanofluid was higher than that of the Cu nanofluid under a magnetic field environment. According to Table 1 above, hybrid nanoparticles also include Al_2_O_3_-TiO_2_, TiO_2_-SiO_2_, MWCNT-SiO_2_, Al_2_O_3_-CuO, Fe_3_O_4_-CQD, CuO-CQD, Al_2_O_3_-Ag, Al_2_O_3_-Cu, Al_2_O_3_-SiC, etc. However, current research on hybrid nanofluid applications in PEFMCs only regard the Al_2_O_3_-SiO_2_ hybrid nanofluid. For example, Johari [179] prepared an Al_2_O_3_-SiO_2_ hybrid nanofluid with different percentages and analyzed its heat-dissipation capacity using an experimental method. The results proved that the thermal conductivity of the Al_2_O_3_-SiO_2_ nanofluid was higher than that of a single nanofluid. Khalid [180] prepared different percentages of Al_2_O_3_-SiO_2_ nanofluids. The experimental results showed that the thermal and electrical conductivity of the Al_2_O_3_-SiO_2_ nanofluid decreased as the percentage of Al_2_O_3_ increased. Meanwhile, the viscosity of the Al_2_O_3_-SiO_2_ nanofluid increased. The test results indicated that the optimal percentage of Al_2_O_3_-SiO_2_ nanofluid as a coolant for PEMFC was 10:90. 

In addition, many researchers have investigated different types of hybrid nanofluids in electric vehicles [181,182,183,184,185]. It has been proved that hybrid nanofluids have become the future research direction of nanofluid applications in fuel cell vehicles. A summary of the single and hybrid nanofluids is presented in Table 5 below.

## 4. The Challenges of Nanofluid Application in Fuel Cell Vehicles

### 4.1. Nanofluid Stability and Cleaning

Research on nanofluid applications in fuel cell vehicles has been focused on the heat-dissipation performances of PEFMCs. Although current research has proven that nanofluids can effectively enhance the heat-dissipation capacity of thermal management systems, the stability problem is inevitably considered when nanofluids are applied in fuel cell vehicles. In addition, nanofluid cleaning is a significant challenge, as nanofluid coolants must be replaced at the end-of-life cycle because nanoparticle sediments accumulate in microchannels and corners.

Nanofluid stability

According to the theory of electrostatic stabilization (Derjaguin Landau Vewey Overbeek), if the van der Waals attraction force is greater than the electrostatic repulsion force, the nanoparticles will collide with each other. This causes the suspended nanoparticles to agglomerate and form clusters. The nanoparticles gradually settle under the influence of gravity. They even block the microchannels and affect the thermal conductivity and viscosity of the nanofluid. From a macroscopic perspective, different influencing factors, such as temperature, concentration, pH, and particle size, lead to destabilization and precipitate formation. Hence, it is necessary to analyze the nanofluid stability from macroscopic and microscopic perspectives.

Nanofluid preparation includes the one-step method and the two-step method. The one-step method involves incorporating nanoparticles directly into the base fluid without intermediate steps, such as nanoparticle drying, storage, and dispersion. This method produces nanofluids with good stability; however, it is only suitable for preparing small quantities of nanofluids because of its high cost. The two-step method refers to the preprocessing of the nanoparticle, which is followed by the mixing of the nanoparticle and the base fluid. This method is suitable for preparing large quantities of nanofluids at a low cost. The stability of this method is not as good as that of the one-step method, and a dispersant is required to maintain good stability [186]. The preparation equipment is illustrated in Figure 11 below.

To solve the nanofluid settlement problem, many researchers have used auxiliary dispersion techniques to improve nanofluid stability. These techniques include physical and chemical dispersions. Physical dispersion involves ball milling, ultrasonic vibration, and magnetic stirring. Ball-milling dispersion refers to the use of planetary ball milling to reduce the average number of clusters. Ultrasonic vibration includes water bath ultrasonic vibration and probe ultrasonic vibration. The latter method is more effective in breaking particle clusters and reducing the average cluster size. A magnetic stirrer can prevent bubble generation during nanofluid preparation by adjusting the stirring speed, which keeps the nanofluid stable for a short time. Chemical dispersion involves the addition of surfactants and pH adjustment. Surfactants can be classified as ionic, nonionic, amphoteric, and complex. The dispersion results are presented in Table 6.

According to the summarization in Table 5, current nanofluid applications in PEMFCs include ZnO nanofluid, Al_2_O_3_ nanofluid, TiO_2_ nanofluid, and Al_2_O_3_-SiO_2_ nanofluid. The stability of these nanofluids have been investigated by some researchers. For example, Choudhary [193] used zeta measurement and visual inspection methods to analyze an Al_2_O_3_ nanofluid’s stability. If the zeta potential was higher, the stability of the Al_2_O_3_ nanofluid was better. Witharana [195] also used zeta measurement, visual inspection, and particle size measurement methods to analyze the stability of a TiO_2_ nanofluid. In addition, the effects of sunlight on the stability of the TiO_2_ nanofluid were assessed. Sunlight caused the TiO_2_ nanofluid’s size to increase up to 45% in three days. The sedimentation velocity is a key parameter to evaluate the stability of a TiO_2_ nanofluid. Adil [196] investigated the effects of a dispersant on the stability of a ZnO nanofluid. Anionic surfactants including SDS, SDBS, and oleic acid were compared. The testing results indicated that using a dispersant, adjusting the pH value, and ultrasonication methods are recommended to increase the stability of a ZnO nanofluid.

Addressing the stability problems of nanofluids is a fundamental requirement for their application in fuel cell vehicles. Although many researchers have elucidated the specific factors affecting nanofluid stability, it is necessary to reveal the hidden laws of these factors. A schematic diagram of nanofluid destabilization and sedimentation is shown in Figure 12.

The main factors influencing the stability of nanofluids include the temperature, concentration, pH, and particle size. (1) Temperature: When the temperature changes, the dielectric constant of the base fluid also changes (i.e., a higher dielectric constant indicates a better stability of the nanofluid). (2) Concentration: The average separation distance of the nanoparticle decreases as the concentration increases. The van der Waals attraction also increases. If the Van der Waals attraction is greater than the electrostatic repulsion, the nanoparticles will agglomerate. Hence, a higher nanofluid concentration implies easier settling. (3) The pH affects the charge density on the nanoparticle surface, decreasing the zeta potential. (4) Particle size: This affects the van der Waals attraction and electric double-layer repulsion. When the nanoparticle size is small, the number of atoms on the nanoparticle’s surface is higher. The density increases in the active position. The tendency for aggregation is enhanced, indicating that the nanofluid settles more easily.

Nanofluid cleaning

According to the literature, the coolant must be replaced every four years in electric vehicles [197]. Because nanoparticle sedimentation accumulates in the microchannels and corners, the coolant is difficult to clean completely. Generally, the coolant inside a vehicle cooling system must be cleaned in two steps. (1) Coolant appearance inspection: If the coolant undergoes sedimentation and produces an abnormal smell near the end of its useful life, it indicates that the coolant has deteriorated. Therefore, the coolant should be replaced as early as possible. (2) pH value inspection: The pH is affected by additives that can effectively protect the metal and rubber seals in the engine. It is necessary to adjust the pH to maintain the anti-corrosion characteristics of the coolant. Finally, the coolant must be changed under cold conditions. Each branch pipe and joint in the cooling system must be checked for leakages. The engine operation must be maintained and the coolant inside the pipes must be cleaned. Finally, a new coolant is injected into the cooling system [198].

However, the problem of nanofluid stability is the greatest challenge that prevents its use in fuel cell vehicles. If nanoparticles accumulate in corners, such as in the flat tubes of a radiator, it is difficult to completely clean them. This affects the thermophysical properties of nanofluid coolants.

### 4.2. Nanofluids in the Cooling Process

#### 4.2.1. Nanofluid Erosion and Abrasion of the Microchannel Surface

Nanofluids are composed of nanoparticles (e.g., metal, metal oxide, nonmetal, nonmetal oxide, and nitrides) and base fluids (e.g., water, oil, and ethylene glycol). During circulation in the cooling loop, the nanoparticles impact the surfaces of the microchannels. In the long term, nanoparticles may wear the microchannels, particularly those in the radiator flat tube and the flow channel of PEMFCs, as shown in Figure 13. The bipolar structure of a PEMFC consists of a membrane electrode, a bipolar plate, and an end plate. The coolant flows through the microchannels of the PEMFC. Heat is generated by the PEMFC and exchanged by the coolant with cold air from the outside. However, during heat exchange, the nanoparticles continuously rub against the surface of the microchannels. In addition, the electrical conductivity of the nanoparticles affects the normal operation of the PEMFC. In the radiator in the cooling system, the nanofluid flows through flat tubes and transfers heat from the inside to the outside. In the flow process, the nanoparticles continuously rub against the surface of the flat tube, which wears the inner surface of the flat tube. The characteristics of the PEMFC and radiators are affected by long-term wear. Hence, it is necessary to reduce the wear effect of the nanofluid as a coolant for PEMFCs and radiators.

Although many researchers have focused on the heat transfer performance of nanofluids, the erosion and abrasion effects of nanofluids on microchannel surfaces have been poorly reported. Eneren [199] analyzed and compared the abrasion, erosion, and corrosion of metallic surfaces and components caused by nanofluids. Abrasion means that hard particles pass over the surface of a solid metal, causing material loss. Erosion refers to the impact of particles in liquids/gases on a solid surface, causing damage to the solid surface. Corrosion refers to the damage of a solid metallic surface via chemical reactions. These effects are a significant challenge for nanofluid applications in fuel cell vehicles, which require extensive experiments and long-term observations.

#### 4.2.2. Pump Transport Power

The viscosity of the nanofluid increases as the nanofluid concentration increases. This causes the pumping power to exponentially increase. Hence, the nanofluid concentration should not infinitely increase. Karami [200] investigated the effects of various factors, such as the volume fraction, duct length, Reynolds number, and tube diameter, on pumping power. The study results proved that a high-volume fraction of the nanofluid was not reasonable due to the cost of the pumping power. Ho [201] analyzed the friction factor, pumping power, average heat transfer coefficient, thermal resistance, and maximum temperature using an Al_2_O_3_/WEG nanofluid as a coolant. The porosity of the nanofluid increased, causing the pumping power to increase. Pandey [202] tested the heat transfer capacity and frictional losses in a heat exchanger. The experimental results also demonstrated that the pumping power increased as the nanofluid concentration increased. However, some researchers have considered that the pumping power can be ignored because the heat transfer capacity considerably increases when using a nanofluid [203,204]. Therefore, it is challenging to explore the influence of nanofluid viscosity on the pump transport power.

#### 4.2.3. Nanofluid Electrical Conductivity

A nanofluid is a mixture of nanoparticles and a base liquid. However, nanoparticles that include Al_2_O_3_, ZnO, and TiO_2_ are metal oxides with electrical conductivity. According to the coolant requirement for PEMFCs, the electrical conductivity should be less than 5 μs/cm. Many researchers have investigated the electrical conductivities of Al_2_O_3_/WEG, ZnO/WEG, and TiO_2_/WEG nanofluids. For example, Khdher [205] investigated the electrical conductivity of an Al_2_O_3_/bio-glycol nanofluid and measured the electrical conductivity using a measurement apparatus. The test results indicated that adding nanoparticles into the base fluid could increase the electrical conductivity to 53–154 μs/cm at 30–80 °C. Subramaniyan [206] analyzed the electrical conductivities of three types of nanofluids: TiO_2_/W, TiO_2_/EG, and TiO_2_/PG. The research results showed that the TiO_2_/W nanofluids had higher dielectric constants than EG and PG at the same temperature and frequency, which could be adjusted by changing the volume fraction of the nanofluid. Chilambarasan [207] selected ZnO/WEG and ZnO/PEG nanofluids as research targets. The electrical conductivity increased when the temperature was increased at all concentrations.

Regarding hybrid nanofluids, Johari [180] investigated an Al_2_O_3_-SiO_2_/bio-glycol (BG) hybrid nanofluid as a coolant for a PEMFC. Compared to conventional ethylene glycol, green bio-glycol (BG) exhibited a better performance than the base fluid. The thermal conductivity also increased. Consequently, the electrical conductivity decreased. This is shown in Figure 14. The single SiO_2_ nanofluid had the highest electrical conductivity at 268.8 μs/cm, but the single Al_2_O_3_ nanofluid’s electrical conductivity was 227.3 μs/cm. The Al_2_O_3_-SiO_2_ hybrid nanofluid’s electrical conductivity was lower than that of the single nanofluid. The electrical conductivity of the other mixture ratio of the Al_2_O_3_-SiO_2_ hybrid nanofluid was approximate to that of the single Al_2_O_3_ nanofluid.

Based on the literature review above, the challenges of nanofluid application in fuel cell vehicles include: (1) nanofluid stability (how to maintain low-term dispersibility of nanoparticles in the base fluid), (2) nanofluid cleaning (how to clean up remaining nanoparticles in microchannels), (3) erosion and abrasion by nanofluids (how to eliminate or reduce erosion and abrasion), (4) nanofluid pump transport power (how to balance a decrease in the heat-dissipation capacity and a decrease in the pumping power), and (5) nanofluid electrical conductivity (how to achieve the requirement of electrical conductivity for coolants in a fuel cell vehicle).

## 5. Research Directions for Nanofluids in the Future

### 5.1. Nanofluid Stability Improvement

Solving nanofluid destabilization and sedimentation problems is a basic requirement for nanofluids’ use in fuel cell vehicles. It is necessary to investigate and analyze the factors influencing nanofluid stability, including the temperature, concentration, pH value, and particle size. However, previous research has only focused on one or two factors, which was difficult to summarize. The nanofluid preparation process includes physical and chemical dispersion techniques, such as ball milling, ultrasonic vibration, magnetic stirring, addition of a surfactant, and adjustment of the pH value. These auxiliary dispersion techniques can effectively improve a nanofluid’s stability. However, it is difficult to summarize the principle of the auxiliary dispersion technique.

### 5.2. Hybrid Nanofluid Application

For oil-fueled vehicles and electric vehicles, researchers have investigated different types of hybrid nanoparticles, such as GnP-CNC, Al_2_O_3_-TiO_2_, TiO_2_-SiO_2_, MWCNT-SiO_2_, CuO-CNT-graphene, Al_2_O_3_-CNT-graphene, Al_2_O_3_-CuO, Fe_3_O_4_-CQD, CuO-CQD, Al_2_O_3_-Ag, Al_2_O_3_-Cu, Al_2_O_3_-SiC, and Al_2_O_3_-CuO, among others. Current research has verified that hybrid nanofluids have better heat transfer capacities than single nanofluids. However, in fuel cell vehicles, only a few researchers have explored the heat-dissipation performance of an Al_2_O_3_-SiO_2_/WEG nanofluid as a coolant. Hence, it would be of considerable scientific value to study more types of hybrid nanofluids for fuel cell vehicles.

### 5.3. Reduction in Erosion and Abrasion by Nanofluids

During the circulation of a nanofluid in a cooling system, the nanoparticles impact and wear the surface of the microchannels, which is particularly true for the flat tube and flow channel of PEMFCs. The characteristics of PEMFCs and radiators are affected by long-term wear. However, few researchers have investigated erosion and abrasion of the surfaces of flat radiator tubes and microchannels inside PEMFCs caused by nanofluids. It is important to analyze the impact phenomenon using numerous experiments and observations from macroscopic and microscopic perspectives. Finally, the nanofluid with the lowest impact on erosion and abrasion must be selected for comparison and analysis.

### 5.4. Thermal Conductivity Model of Nanofluids

The thermal conductivity of a nanofluid is an important thermophysical property. Many researchers have proposed classic thermal conductivity models, such as the Maxwell, Braggeman, Wasp, Hamilton and Crosser, and Yamada and Ota models. There is no unified theoretical model to accurately calculate nanofluid thermal conductivity. A few researchers have proposed a theory on thermal conductivity of the adsorption layer, which refers to the liquid layer formed around solid nanoparticles between the base fluid and the nanoparticles. This reflects the internal heat transfer mechanism of the nanofluid more accurately. Hence, it would be of great scientific value to explore the thermal conductivity of nanofluids using the adsorption layer theory.

### 5.5. Electrical Conductivity of Nanofluids

According to current research, nanofluid applications in PEMFC include Al_2_O_3_/WEG, ZnO/WEG, and TiO_2_/WEG, which are metal oxides with electrical conductivity. Based on the requirements for coolants for PEMFCs, the electrical conductivity should be less than 5 μs/cm. When nanofluids flow through a PEMFC microchannel, they inevitably contact the inner surface of the BP. If a nanofluid has a high electrical conductivity, it can cause electric leakage inside the PEMFC, affecting the output power of PEMFC. Hence, exploring nanofluids with low electrical conductivities in the future is necessary.

## 6. Conclusions

With the development of high-power fuel cell stacks, heat-dissipation requirements have become more stringent. Reinforcing the heat transfer performance of the coolant is simpler, has a lower cost, and increases the rotation speed than structural optimization. Nanofluids are a type of heat-dissipation-enhancement coolants that have been applied in different industrial fields. This study summarized the challenges and research directions of nanofluids as coolants in fuel cell vehicles. The main conclusions of this study are as follows.

(1) Nanofluids have great potential as coolants in oil-fueled, pure electric, and fuel cell vehicles. Many researchers have verified this and have highlighted their disadvantages. Nanofluids can be used as coolants, refrigerating fluids, and heat transfer media in oil-fueled vehicle cooling systems, air conditioner systems, and exhaust power generation systems. In pure electric vehicles, nanofluids can be considered as coolants in battery-cooling systems. However, for fuel cell vehicles, the current research has only focused on the heat-dissipation performances of nanofluids in PEMFCs.

(2) The thermal management system of an FCV comprises heat dissipation and air conditioning systems. Heat-dissipation systems can be classified into heat-generation and heat-dissipation systems. Based on the analyses and comparisons performed in numerous studies, the heat transfer capacity of a coolant is better than that of the heat-dissipation system. This further proves that nanofluids have considerable potential in replacing existing coolants.

(3) The thermophysical properties of nanofluids, including the density, specific heat, thermal conductivity, and convective heat transfer coefficient, have been extensively studies. The equations for the thermophysical properties have been analyzed in detail. ZnO/WEG, Al_2_O_3_/WEG, TiO_2_/WEG, and BN/WEG nanofluids were analyzed as coolants in PEMFCs. In this study, four types of nanofluids were selected along with their density, specific heat, thermal conductivity, and convective heat transfer coefficient. Moreover, this study mainly analyzed the thermal conductivity of the nanofluid and the Al_2_O_3_-SiO_2_/WEG nanofluid.

(4) According to a literature review, there are some challenges in using nanofluids as coolants in fuel cell vehicles, including nanofluid stability, cleaning, erosion and abrasion, pump transport power, and electrical conductivity. Among these, nanofluid stability is the biggest obstacle that prevents its use in fuel cell vehicles. Hence, it is necessary to improve the nanofluid preparation methods, which affect the stability of the nanofluid, from macroscopic and microscopic perspectives.

(5) Finally, this study proposed five research directions for using nanofluids as coolants in fuel cell vehicles in the future. They include nanofluid stability improvements, hybrid nanofluid applications, reductions in erosion and abrasion by nanofluids, a nanofluid thermal conductivity model, and nanofluid electrical conductivity.

## Figures and Tables

**Figure 1 nanomaterials-13-02861-f001:**
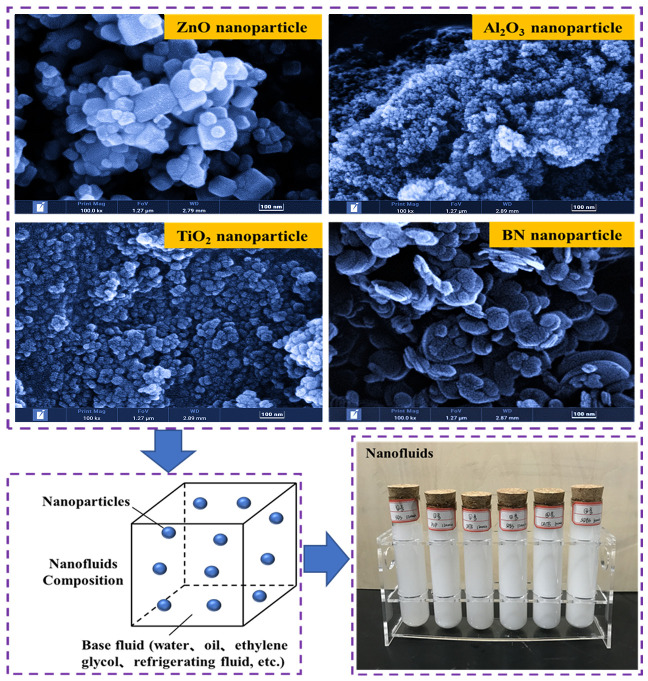
Schematic diagram of nanofluid compositions.

**Figure 2 nanomaterials-13-02861-f002:**
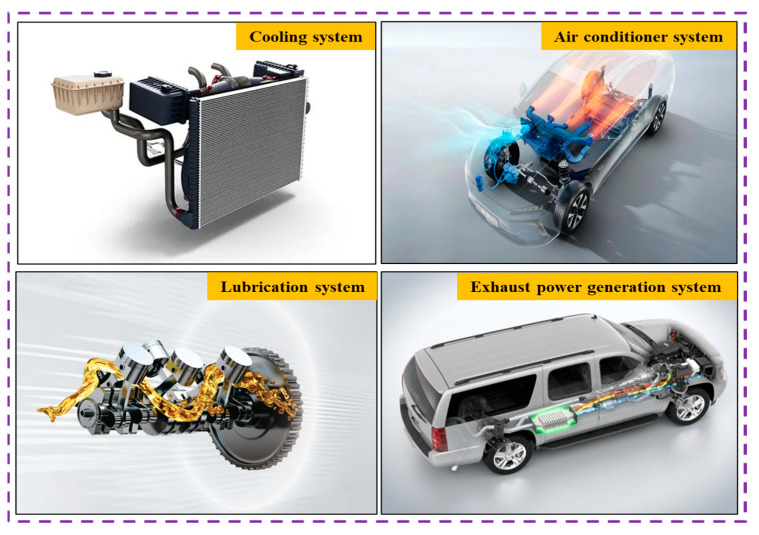
Nanofluid applications in oil-fueled vehicles.

**Figure 3 nanomaterials-13-02861-f003:**
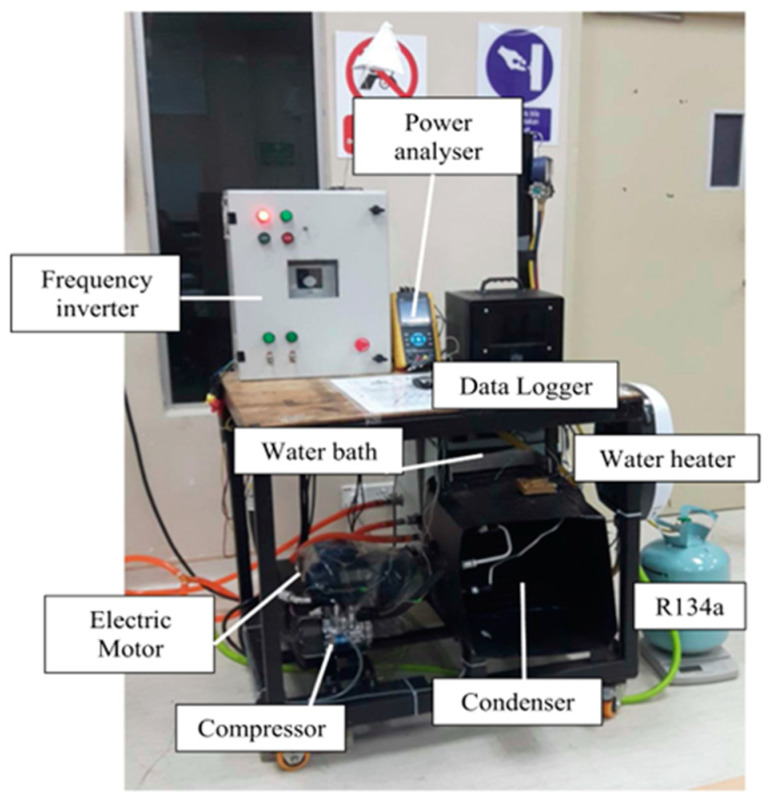
AAC system experimental test bench [110].

**Figure 4 nanomaterials-13-02861-f004:**
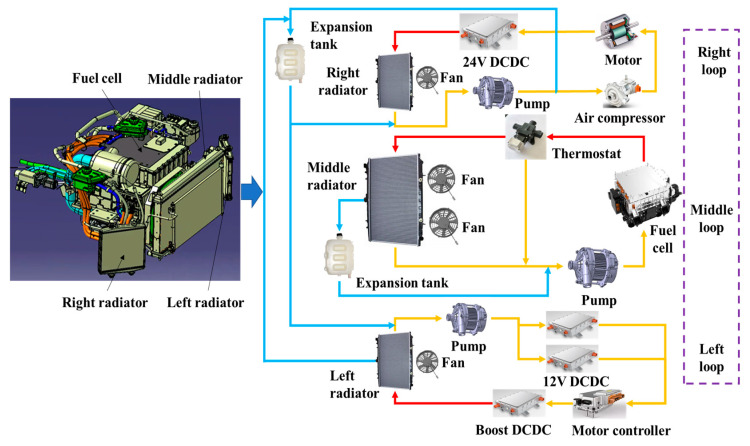
Thermal management system of a fuel cell vehicle.

**Figure 5 nanomaterials-13-02861-f005:**
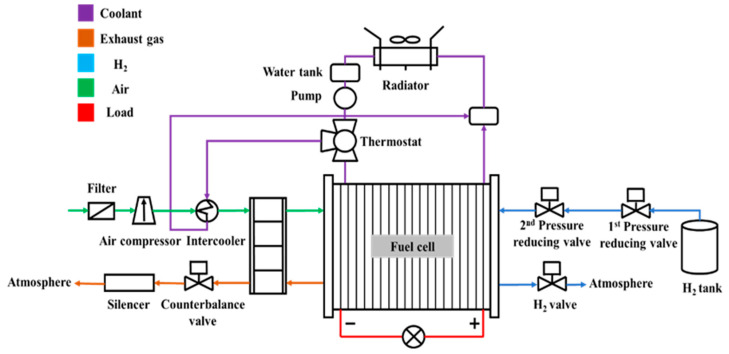
PEMFC engine [139].

**Figure 6 nanomaterials-13-02861-f006:**
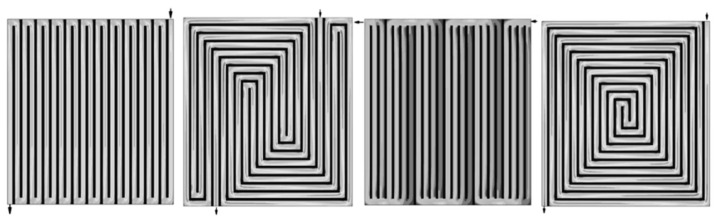
Different flow channel structures inside a fuel cell [141].

**Figure 7 nanomaterials-13-02861-f007:**
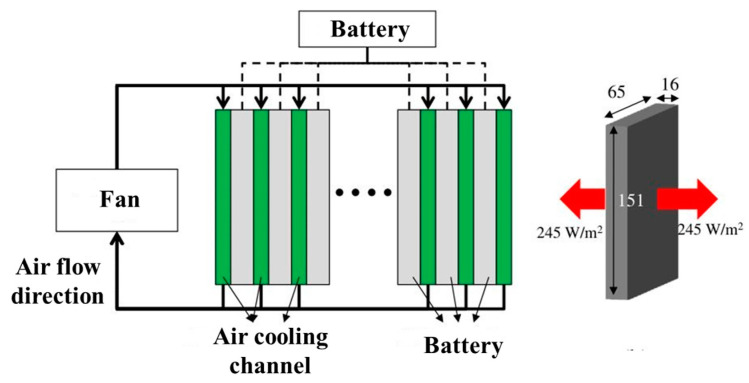
Schematic diagram of forced air cooling [146].

**Figure 8 nanomaterials-13-02861-f008:**
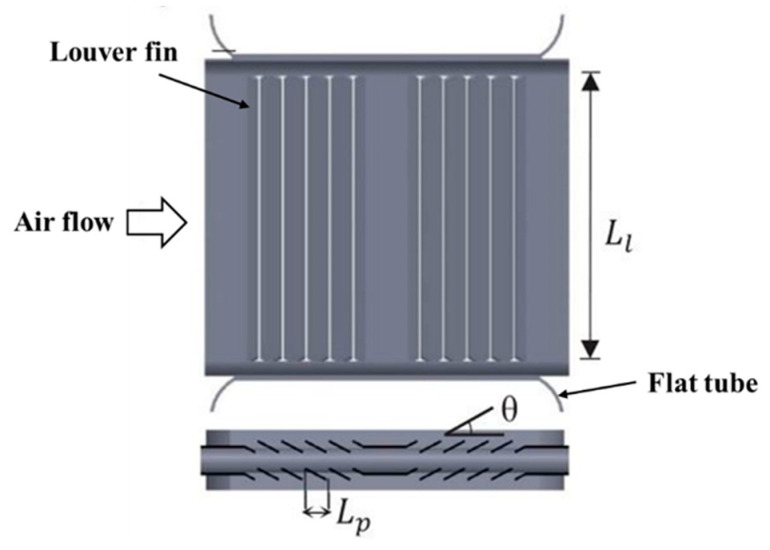
Structure of louver fin radiator [151].

**Figure 9 nanomaterials-13-02861-f009:**
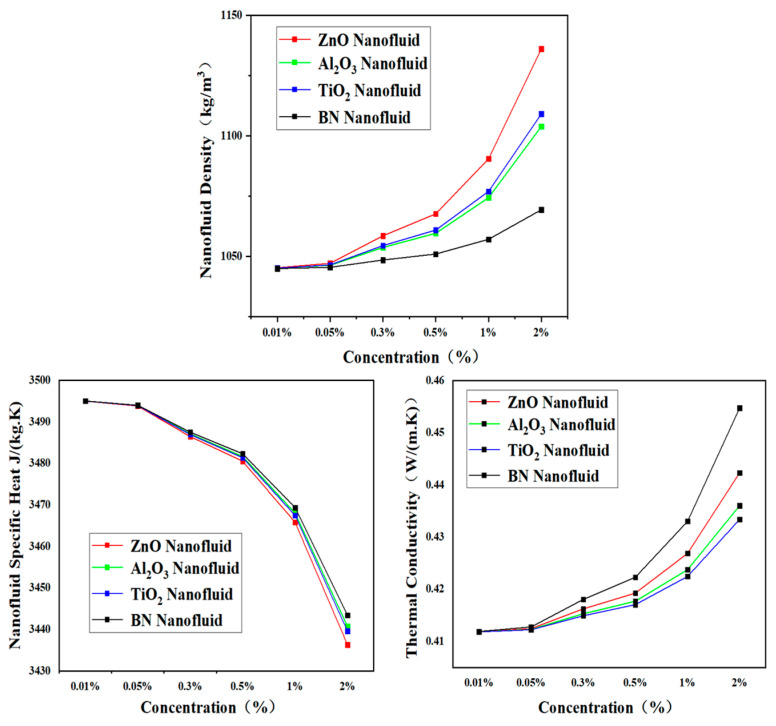
Thermophysical properties of four types of nanofluids.

**Figure 10 nanomaterials-13-02861-f010:**
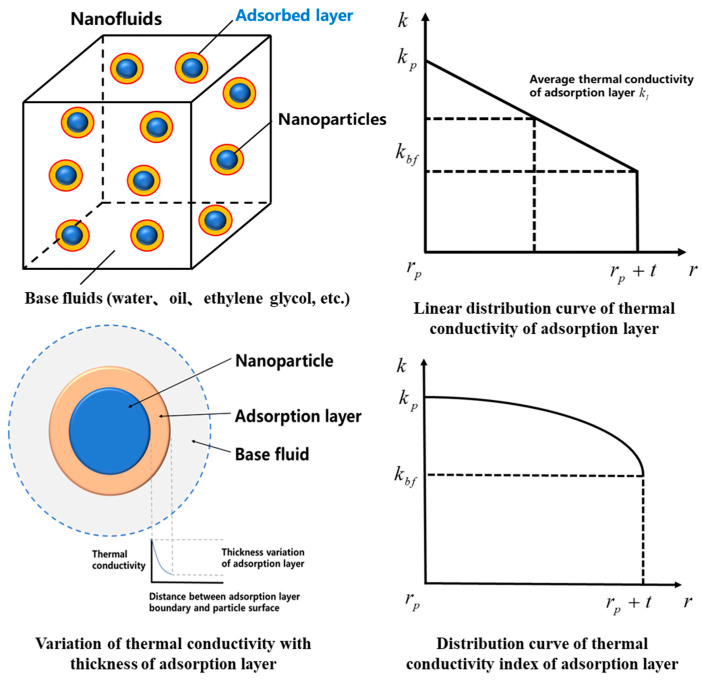
Schematic diagram of nanoparticles and adsorption layers.

**Figure 11 nanomaterials-13-02861-f011:**
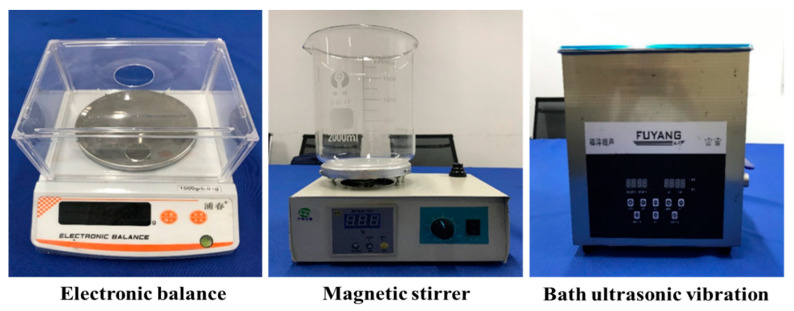
Nanofluid preparation equipment.

**Figure 12 nanomaterials-13-02861-f012:**
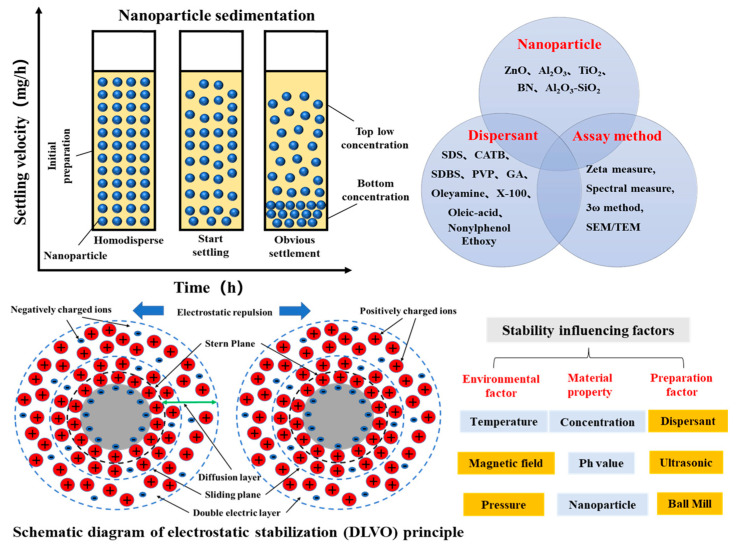
Schematic diagram of nanofluid destabilization and sedimentation [194].

**Figure 13 nanomaterials-13-02861-f013:**
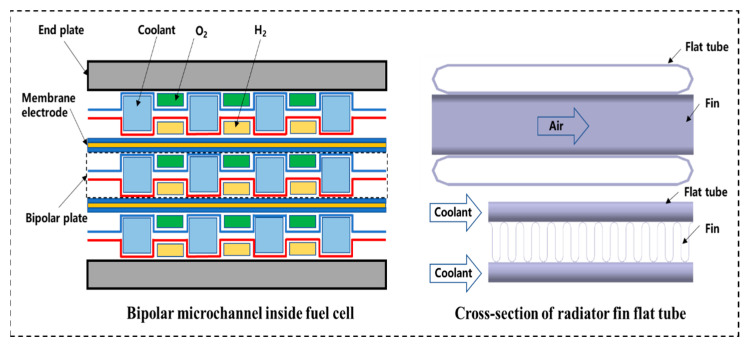
Schematic diagram of microchannels in PFMFC and radiator.

**Figure 14 nanomaterials-13-02861-f014:**
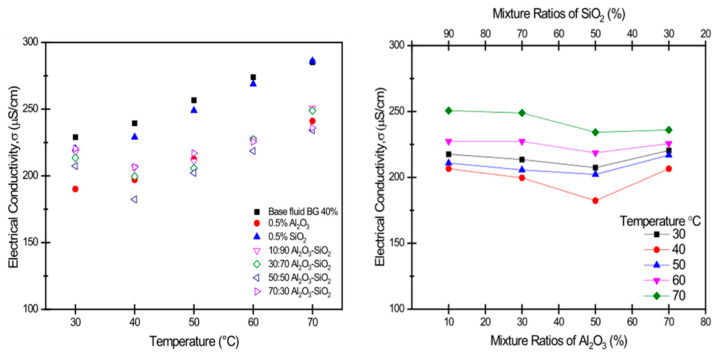
Electrical conductivity of Al_2_O_3_-SiO_2_/BG hybrid nanofluid [206].

**Table 1 nanomaterials-13-02861-t001:** Nanofluid applications in oil-fueled vehicles’ cooling systems [53,54,55,56,57,58,59,60,61,62,63,64,65,66,67,68,69,70,71,72,73,74,75,76,77,78,79,80,81,82,83,84,85,86,87,88,89,90,91,92,93,94,95,96,97,98,99].

Nanoparticle Type	Base Fluid	Concentration	Quantitative Analysis	Author/Ref.
GnP, CNC	WEG	0.2%	OHTC increased by 46.72%	Yaw [53]
TiO_2_	Water	0.1%, 0.2%, 0.3%	Effectiveness of car radiator increased by 47% with 0.2% TiO_2_	Ahmed [54]
TiO_2_	Water	0.025%, 0.05%, 0.1%, 0.2%	Heat transfer rate increased by 22.2% with 0.05% TiO_2_	Elibol [55]
Al_2_O_3_, TiO_2_	WEG	0.3%	Thermal performance increased by 24.21% with 0.3% Al_2_O_3_	Said [56]
TiO_2_	Water	0.1–4%	Nunf/Nubf≈1 with low concentration	Elibol [57]
Fly ash	WEG	0.2–2%	Heat transfer rate increased by 7 kW	Palaniappan [58]
Al_2_O_3_	Water	0.1%, 0.2%, 0.4%	Heat transfer coefficient increased by 55% with 0.4% Al_2_O_3_	Yasuri [59]
TiO_2_	WEG	0.1%, 0.3%, 0.5%	Heat transfer rate increased by 37%	Arora [60]
Al_2_O_3_	Water	0.1%, 0.5%, 1%, 1.5%, 2%	Heat transfer performance was the optimum at 1%	Ali [61]
CuO	Water	0.2–0.5%	Pressure drops increased by 20% with 0.5% CuO	Sokhal [62]
Al_2_O_3_	Water	0.5–3%	Irreversibility increased by 0.3% at 15 LPM	Kumar [63]
Al_2_O_3_	EG	0.08%, 0.5%, 1%	Thermal performance increased up to 5%	Goudarzi [64]
Al_2_O_3_	WEG	0.3%, 0.6%, 0.9%, 1.2%	Heat transfer coefficient increased by 9% with Al_2_O_3_	Sheikhzadeh [65]
TiO_2_, SiO_2_	Water	1%, 1.5%, 2%	Maximum Nusselt number increased by 11% and 22.5%	Hussein [66]
CuO	WEG	0.05–0.8%	Heat transfer coefficient increased by 55%	Heris [67]
MWCNT, SiO_2_	WEG	0.1%	Cooling power increased by 40%	Kumar [68]
CuO	Water	0–0.4%	Overall heat transfer coefficient increased by 8%	Naraki [69]
Al_2_O_3_	WEG	0.2%, 0.6%, 1%	Temperature effect was 41.72%	Seraj [70]
Zn, ZnO	Water	0.15%, 0.25%, 0.5%	Radiator area reduced by 24%	Sonage [71]
CuO, CNT	Water	1–3%	Heat transfer rate and efficiency increased by 19.35% and 7.2%	Sahoo [72]
Al_2_O_3_, CNT	Water	1–3%	Irreversibility and entropy change increased by 42.45% and 27.27%	Sahoo [73]
Al_2_O_3_, CuO	WEG	1–10%, 1–6%	Average heat transfer coefficient increased by 94% and 89%	Vajjha [74]
Graphene oxide	WEG	0.1%	Maximum convective heat transfer increased by 69.7%	Shankara [75]
Al_2_O_3_	W/WEG	0–1%	Highest Nusselt number enhancement up to 40%	Peyghambarzadeh [76]
TiO_2_	Water	1%, 2%, 3%, 4%	Heat transfer efficiency increased by 20%	Hussein [77]
Al_2_O_3_, CuO	Water	1%, 3%, 5%, 7%	Heat transfer coefficient increased by 45% and 38%	Elsebay [78]
Graphite	WEG	0.6%, 1%	Overall heat transfer coefficient increased by 11.7%	Akash [79]
CuO	WEG	0–2%	Air frontal area was reduced by 18.7%	Leong [80]
Al_2_O_3_	WEG	0.25%, 0.5%, 1%	Heat transfer performance increased by 37.2%	Karagoz [81]
Al_2_O_3_	Water	0.1–1%	Heat transfer rate increased by 45%	Peyghambarzadeh [82]
Graphene	WEG	0.1–0.5%	Convective heat transfer coefficient increased by 51%	Selvam [83]
Fe_3_O_4_, CuO, Al_2_O_3_, SiO_2_	WEG	0.1%, 0.3%, 0.7%, 1%	Heat transfer efficiency increased by 3.2–45.9%	Yıldız [84]
Al_2_O_3_	WEG	0–1%	Highest thermal conductivity increased by 8.3%	Elias [85]
TiO_2_, SiO_2_	Water	1–2.5%	Effectiveness increased by 24% and 29.5%	Hussein [86]
Fe_3_O_4_-CQD, CuO-CQD	Water	0.5%	Effectiveness increased by 12% and 25%	Mousavi [87]
SiO_2_	Water	0.04%, 0.08%, 0.12%	Heat transfer rate max. increased by 36.92%	Shah [88]
CuO, Fe_2_O_3_	Water	0.15%, 0.4%, 0.65%	Overall heat transfer coefficient max. increased by 9%	Peyghambarzadeh [89]
Fe_3_O_4_	Water	0–0.9%	Radiator heat transfer performance increased by 21%	Tafakhori [90]
Al_2_O_3_	WMEG	0.2–0.8%	Heat transfer rate increased by 30%	Subhedar [91]
MWCNT	WEG	0.025%, 0.05%, 0.1%	Heat transfer rate and OHTC increased by 4.6% and 4.4%	Contreras [92]
CuO, TiO_2_, Al_2_O_3_	Water	-	K_nf_ was 0.72 with 5% CuO nanofluid	Khan [93]
Al_2_O_3_	WEG	0–1%	Nu was 237% higher than WEG	Delavari [94]
Al_2_O_3_(Ag, Cu, SiC, CuO)	WEG	0–1%	Cooling flow rate reduced by 3.1%	Sahoo [95]
SiC	WEG	0.5%	Thermal conductivity max. increased by 53.81%	Li [96]
MWCNTs	SG	0.2%, 0.4%, 0.6%	Nusselt number max. increased by 18.39%	Sivalingam [97]
MWCNTs	WEG	0.1%, 0.25%, 0.5%	Average heat transfer coefficient max. increased by 196.3%	Mhamed [98]
ZnO	WEG	0.01–0.04%	Heat transfer rate max. increased by 36%	Khan [99]

**Table 2 nanomaterials-13-02861-t002:** Cooling methods for fuel cells [140].

Cooling Method	Output Power
Cathode air cooling	<100 W
Separate airflow cooling	200–2000 W
Phase change	1000 W
Liquid cooling	>10 kW

**Table 3 nanomaterials-13-02861-t003:** Heat-dissipation method and improvement method.

Component	Heat-Dissipation Method	Improvement Method
Fuel cell	Liquid cooling	Structure
		Control strategy
		Coolant
Power battery	Air/liquid/phase change/heat pipe cooling	Structure
		Coolant
Motor and controller	Natural/air/liquid/oil cooling	Structure
		Coolant
Air compressor	Liquid-cooled/air-cooled	Structure
		Coolant
DC/DC converter	Liquid cooling	Structure
		Coolant

**Table 4 nanomaterials-13-02861-t004:** Nanoparticles’ density, specific heat, and thermal conductivity.

Type	Density kg/m^3^	Specific Heat J/(kg·K)	Thermal Conductivity W/(m·K)
ZnO	5610	544	25
Al_2_O_3_	4000	765	36
TiO_2_	4260	710	8.2
BN	2270	900	260

**Table 5 nanomaterials-13-02861-t005:** Single and hybrid nanofluids in PEFMCs [133,134,135,179,180].

Type	Author	Method	Result
ZnO	Islam	Experiment	Radiator area reduced by 27%
Al_2_O_3_	Zakaria	Experiment	Heat dissipation increased by 13.87%
TiO_2_	Islam	Experiment/Theory	Thermal conductivity increased by 10%
Al_2_O_3_-SiO_2_	Johari	Experiment	Thermal conductivity increased by 21%
Al_2_O_3_-SiO_2_	Khalid	Experiment	10:90 was the most feasible ratio

**Table 6 nanomaterials-13-02861-t006:** The research on nanofluid stability [187,188,189,190,191,192,193,194].

Author	Nanoparticle	Base Fluid	Surfactant	Stability
Yu [187]	MWCNT	Water	SDS	90 days
Islam [188]	SWCNT	Water	SDBS	90 days
Tang [189]	MWCNT	Water	PVP	60 days
Li [190]	Cu	Water	CATB	7 days
Hwang [191]	Ag	Oil	Oleic acid	60 days
Wu [192]	SWCNT	Water	Humic acid	10 days
Choudhury [193]	Al_2_O_3_	Water	SDS	16 days
Mo [194]	TiO_2_	Water	SDS	12 days

## Data Availability

No new data were created or analyzed in this study. Data sharing is not applicable to this article.

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
