# Peer review of "A Review of Nanofluids as Coolants for Thermal Management Systems in Fuel Cell Vehicles"

_nanomaterials, 2023, doi:10.3390/nano13212861_

Round 1

Reviewer 1 Report

Comments and Suggestions for Authors

Comments on the Quality of English Language

  Language: The English must be revised by a native speaker and British English or American English should be used throughout the manuscript.

Author Response

       Thank you very much for your comments concerning our manuscript. Those comments are all valuable and very helpful for revising and improving our paper. We have studied comments carefully and have made correction point by point  which can be seen in attachment file. Revised portion are marked in red or blue in the manuscript.

Reviewer 2 Report

Comments and Suggestions for Authors

The review is well written and deserves pubblication.

As a general consideration I can say that from my point of you the review article is usefull for the scientific community and deserve pubblication.   I can add a few extra comments:

  1. The main of the paper is focused on nanofluid to be used as coolant into fuel cells: being a review it illustrates the latest improvements in the filed
  2. It is a review so the novelty parameter does not apply 
  3. The review is focused on the nanofluid field
  4. The authors illustrate the work on the field, they do not apply any specific methodology.
  5. Conclusions are in my opinion appropriate
  6. References are appropriate
  7. I strongly suggest to consider a mother toungue for the language and the clarity of the text and to take into account the pint highlighted  by reviewer 2

Comments on the Quality of English Language

I suggest a moderate english revision.

Author Response

     Thank you very much for your comments concerning our manuscript. Those comments are all valuable and very helpful for revising and improving our paper. We have studied comments carefully and have made correction point by point which can be seen in the attachment file. Revised portion are marked in red in the manuscript.

Reviewer 3 Report

Comments and Suggestions for Authors

The authors present a literature review for nanofluids as a coolant in fuel cell vehicles. 

The authors should include several other publications in the literature search to make it more comprehensive. 

They should also elaborate more on the future research.

Literature to be added:

Frank et al. Heat Transfer across a fractal surface, Journal of Chemical Physics, 151, 134705, 2019; Solid-like heat transfer in confined liquids, Microfluidics and Nanofluidics, 21, 9, 6 p., 148, 2017; 

Ghorbanian et al. (2016) Scale effects in nano-channel liquid

flows. Microfluid Nanofluid 20(8):121

Papanikolalou et al. Nanoflow over a fractal surface, Physics of Fluids, 28(8), 082001, 2016.

Liu et al. (2005) Dynamics and density profile of water in nanotubes as one-dimensional fluid. Langmuir 21:12025–12030

McGaughey et al. (2004) Thermal conductivity decomposition

and analysis using molecular dynamics simulations. Part I.

Lennard-Jones argon. Int J Heat Mass Transf 47:1783–1798

Sun et al. (2002) Confined liquid: simultaneous observation of a molecularly layered structure and hydrodynamic slip. J Chem Phys 117(22):10311–10314

Comments on the Quality of English Language

Moderate changes

Author Response

(The authors gave the same response as above.)

Reviewer 4 Report

Comments and Suggestions for Authors

The peer-reviewed paper contains a literature review on nanofluid application for heat removing and analyses the application of a nanofluid as coolant for fuel cell vehicles.  It should be noted that using nanoparticle additives to a cooler to intensify heat transfer processes has been discussed in a large number of papers and reviews, but selection of the composition of  nanofluid and properties of the nanoparticles remains a popular topic. The main attention in the reviewed paper is paid to possible areas of application the nanofluids; less attention is paid to the thermophysical properties of nanofluids. This is a shortcoming of the paper submitted for publication in the Nanomaterials. However, the review is of some interest to readers and may be published in the Nanomaterials according to the comment below.

Comment.

1.  The maximal value for the increase in heat transfer coefficient in case nanofluid for the data presented in Fig. 9 does not exceed 2.5 percent. Therefore, the data given in Table 1 which show the increasing of heat transfer by 45 percent and more need to be discussed. Perhaps Table 1 needs to be expanded to include the data with less enhancement of heat transfer, which will correspond to the Fig. 9.

Author Response

(The authors gave the same response as above.)

Reviewer 5 Report

Comments and Suggestions for Authors

The concerned topic is interesting and important, however, the structure of the article is not reader-friendly because it contains a lot of sections and subsections, which are often short and cover the topic in a cursory manner, for example, section 3.3 where the authors analyzed only two papers and cite the other five collectively in a table. Moreover, the manuscript has other flaws, which should be considered before publication:

1. Authors should get permission to reuse figures 3,5,6,7,8 and 14, as well as add to the caption the license number. Simply quoting the source publication is not sufficient.

2. Section 2 seems superfluous since it only deals with structural solutions and does not indicate their impact/relation to the use of nanofluids. I suggest removing it or clearly showing the reference to nanofluids.

3. In section 4.1 authors refer to studies concerning highly conductive nanoparticles, mostly those carbon-based, while such types of nanofluids should not be used in PEMFCs. Here authors should pay more attention to the stability of nanofluids with lower electrical conductivity.

4. The authors should also discuss the Thermo-electrical conductivity ratio (TEC) proposed by Zakaria et al. (10.1016/j.icheatmasstransfer.2014.12.015) and related studies should be discussed in this context.

Author Response

     Thank you very much for your comments concerning our manuscript. Those comments are all valuable and very helpful for revising and improving our paper. We have studied comments carefully and have made correction point by point which can be seen in the attachment file. Revised portion are marked in red or blue in the manuscript.

Round 2

Reviewer 1 Report

Comments and Suggestions for Authors

I believe that the manuscript has been improved and that all issues previously raised have been properly addressed.

Author Response

We really appreciate your comments and suggestion concerning our manuscript. Those comments are all valuable and very helpful for revising and improving our paper. We will keep improving our manuscript based on journal’s publication requirements.

Reviewer 3 Report

Comments and Suggestions for Authors

The authors have cited the new references incorrectly. They should add all the authors' names and publication details, for example, 

M. Frank, M. Papanikolaou, D. Drikakis, K. Salonitis, Heat Transfer across a fractal surface, Journal of Chemical Physics, 151, 134705, 2019

M. Papanikolalou, M. Frank, D. Drikakis, Nanoflow over a fractal surface, Physics of Fluids, 28(8), 082001, 2016.

Comments on the Quality of English Language

None

Author Response

Thank you very much for your comments concerning our manuscript. Those comments are all valuable and very helpful for revising and improving our paper. We have studied comments carefully and have made correction point by point as summarized below. Revised portion are marked in blue in the manuscript.

Reviewer 5 Report

Comments and Suggestions for Authors

The authors addressed most of the comments satisfactorily but left out the issue of obtaining permission from other publishers to use previously published figures. I believe that the issue of respecting copyright and intellectual property is important in the process of producing science.

Author Response

Thank you very much for your comments concerning our manuscript. Those comments are all valuable and very helpful for revising and improving our paper. Due to our negligence, we did not mention the copyright issue in the first revision of our replies. In fact, we had already obtained the permissions of Figure 3,6,7,8,12,14 and Table 2 before submitting the paper for the first submission. These permissions can be seen in the attachment file. We fully agree with your viewpoint about the issue of respecting copyright and intellectual is import in the process of producing science.
